# Enhancing Error Detection on Medical Knowledge Graphs via Intrinsic Label

**DOI:** 10.3390/bioengineering11030225

**Published:** 2024-02-27

**Authors:** Guangya Yu, Qi Ye, Tong Ruan

**Affiliations:** 1Zhejiang Laboratory, Hangzhou 311121, China; y80230105@mail.ecust.edu.cn; 2School of Information Science and Technology, East China University of Science and Technology, Shanghai 200237, China; ruantong@ecust.edu.cn

**Keywords:** medical knowledge graph, error detection, confidence score, graph attention network

## Abstract

The construction of medical knowledge graphs (MKGs) is steadily progressing from manual to automatic methods, which inevitably introduce noise, which could impair the performance of downstream healthcare applications. Existing error detection approaches depend on the topological structure and external labels of entities in MKGs to improve their quality. Nevertheless, due to the cost of manual annotation and imperfect automatic algorithms, precise entity labels in MKGs cannot be readily obtained. To address these issues, we propose an approach named Enhancing error detection on Medical knowledge graphs via intrinsic labEL (EMKGEL). Considering the absence of hyper-view KG, we establish a hyper-view KG and a triplet-level KG for implicit label information and neighborhood information, respectively. Inspired by the success of graph attention networks (GATs), we introduce the hyper-view GAT to incorporate label messages and neighborhood information into representation learning. We leverage a confidence score that combines local and global trustworthiness to estimate the triplets. To validate the effectiveness of our approach, we conducted experiments on three publicly available MKGs, namely PharmKG-8k, DiseaseKG, and DiaKG. Compared with the baseline models, the Precision@K value improved by 0.7%, 6.1%, and 3.6%, respectively, on these datasets. Furthermore, our method empirically showed that it significantly outperformed the baseline on a general knowledge graph, Nell-995.

## 1. Introduction

A knowledge graph (KG) consists of semantic edges and diverse entities, represented as (h,r,t). KGs can be general, such as YAGO [1] and DBpedia [2], or domain-specific, such as biomedical KG [3] and financial KG [4]. Medical KGs, in particular, exhibit unique structures, contain abundant semantic information [5], and play a crucial role in various healthcare applications, including disease diagnosis [6], drug analysis [7], clinical education [8], and data quality management [9]. However, constructing a medical KG from scratch can be a time-consuming and labor-intensive procedure [10]. Recently, there have been studies that propose constructing medical KGs by either fusing existing KGs [11] or extracting entities related to ICD-9 from Freebase [12]. These approaches aim to streamline the process of constructing medical KGs by leveraging existing resources and extracting relevant entities. However, these methods inevitably introduce noisy triplets due to human errors and imperfect algorithms. For example, the correct triplet (Bill Gates, found, Microsoft) might be mistakenly identified as (Bill Gates, found, Google). This can lead to substantial errors in downstream tasks and applications that rely on the accuracy of the knowledge graph [13].

Traditional knowledge graph representation learning algorithms, such as TransE [14], DistMult [15], and RotatE [16], assume the correctness of all triplets in the KG. However, the absence of error detection mechanisms exposes downstream tasks to significant risks. Therefore, it is crucial to develop an error detection algorithm to ensure the reliability of a medical KG.

As the errors present in KGs can be diverse and their nature may be unknown [17], it is nontrivial to detect noisy triplets in KGs [18]. Recently, some studies have enabled error-aware learning against the noisy triplets [19,20]. In particular, CKRL [19] estimates the triplet with confidence through path and KG embedding. To classify the triplets directly with trustworthiness, KGTtm [21] proposes a model that integrates entity, relationship, and KG global-level information. The state-of-the-art error detection method, CAGED [22], effectively combines KG embedding with contrastive learning. This approach estimates the confidence of triplets both locally and globally, enabling the accurate detection of errors in the KG. By utilizing the additional entities’ attribute information, AKAE [23] enhances the error detection process and improves the accuracy of identifying and addressing nontrivial errors in the KG.

While AKAE has shown promising results, obtaining external attribute information for entities can be challenging. Therefore, we propose an approach that leverages the topological structure of the graph and the rich potential entity information within MKGs to extract intrinsic label information. This approach is more versatile and can be applied to various medical KGs without relying on explicit attributes. To this end, we propose the framework named Enhancing error detection on Medical knowledge graphs via intrinsic labEL (EMKGEL). Our contributions are as follows:Noting the abundance of entities’ labels in medical KGs, we propose a novel method that extracts the intrinsic label information of entities via a hyper-view KG. Further, we establish the hyper-view KG ourselves due to its absence;Aiming to integrate the topological information and intrinsic label information, we propose a hyper-view GAT consisting of a bi-LSTM layer for capturing local structural messages and a modified graph attention mechanism for modeling neighborhood information with potential labels’ messages;Ranking the triplet by confidence score, we conduct comprehensive experiments on three medical KGs and a general KG and outperform other methods.

This research aims to make a valuable contribution to the advancement of medical knowledge graphs within the open-source community. The proposed approach can be effectively utilized for identifying errors in knowledge graph construction. By leveraging this method, it is possible to reduce the time required for construction while simultaneously improving the overall quality of the knowledge graph. In summary, the proposed method represents a promising tool for error detection in knowledge graphs and has wide-ranging implications for advancing the field.

## 2. Related Work

The related work of the error-detection research community can be divided into three categories: knowledge graph representation learning, error-aware knowledge graph embedding, and knowledge graph error detection.

### 2.1. Knowledge Graph Representation Learning

Numerous studies have been dedicated to the field of knowledge graph (KG) representation learning, which has emerged as a crucial foundation for various downstream tasks [24,25]. Based on the scoring function used and whether a global graph structure is utilized, KG representation learning methods can be broadly categorized into three groups: translation-based, semantic matching, and GNN-based models.

Translation-based models, exemplified by TransE [14], have garnered widespread attention due to their pioneering work in transforming the KG into a continuous vector space. However, TransE is not well-suited for handling 1-N, N-1, and N-N relations within the knowledge graph. To tackle the issue, several variants of TransE are proposed, including TransH [26], TransR [27], and TransD [28]. In particular, to fit the various relation patterns, including symmetry/antisymmetry, inversion, and composition, RotatE [16] projects the entities and relations into complex vector space via rotation.

Semantic matching models employ a similarity-based scoring function to evaluate the likelihood of the facts. One representative model in this category is RESCAL [29], which utilizes a matrix of full rank to depict relations. Building upon this, DistMult [15] transforms the relation matrix into a diagonal matrix. CompIEx [30] expands on DisMult by incorporating complex space and effectively addressing both symmetry and asymmetry concerns.

GNN-based models integrate neighborhood information to improve the performance of the KG embedding, e.g., ConvKB [31] and CompGCN [32]. Specifically, ConvE [33] utilizes 2D convolution to represent entities and relations. R-GCN [34] obtains the structural information of multi-relation graphs through relation matrices and node neighbors, but the parameters explode as the relations grow. These methods assign the same importance to nodes’ neighbors, while the different nodes make different contributions. To tackle the issue, KGAT [35] introduces an attention mechanism to learn the weight of nodes in a neighborhood.

However, these methods ignore the noisy triplets in KGs, which may decrease the quality of KG representation learning. Thus, it is urgent to develop an algorithm for error-aware KG embedding or to detect errors in the KG.

### 2.2. Error-Aware Knowledge Graph Embedding

Error-aware KG embedding is end-to-end representation learning that learns the embedding of entities and relations with noise. Aiming to detect possible errors, Dong [36] constructed a knowledge vault by fusing existing KG embeddings with probability scores. Recently, CKRL [19] offers a novel approach that incorporates the confidence score while learning knowledge representations in the KG with noise. While CKRL has shown effectiveness in noise detection by confidence scores, its utilization of uniform negative sampling methods and a strict triple quality function can potentially lead to zero loss problems and false detection. To address these challenges, NKRL [20] introduces the concept of negative confidence and proposes a negative sampling method for training. In addition, TransT [37] measures the confidence score via external entity type and description. These models generate confidence scores in representation learning to estimate the triplets, thereby enhancing the robustness of KG embedding with noise.

### 2.3. Knowledge Graph Error Detection

Error detection has been a long-standing challenge for many years. Primarily, studies focused on rule-based methods [38], cluster-based methods [39], and distance-based methods [40]. Among these methods, there have been several innovative works for error detection. Undoubtedly, it is straightforward to rank triplets by their embedding score function, e.g., TransE’s score function is −∥h+r−t∥, where the larger the score, the more reliable the triplet. In addition, a classification-based method, KGClean [41], utilizes a novel method called AL-detect to identify errors. Recently, Jia [21] proposed a triple trustworthiness measurement model that combines intra-triplet, structural, and global inference information. This approach enables the fusion of entity, relationship, and path-level information. The state-of-the-art method, CAGED [22], introduces the concept of a link pattern for data augmentation and subsequently integrates KG embedding and contrastive learning to combine local and global information. Based on this, AKAE [23] incorporates the external entity attribute information.

## 3. Problem Statement

Given a knowledge graph G, we define G=(E,R,S), where E,R,S represent the sets of entity, relations, and triplets, respectively. The major notations used in this paper can be followed in Table 1

**Definition** **1.**
***Errors in KGs.** Considering a triplet (h, r, t), there is an error if there is a mismatch of the head, relation, or tail, represented as (h’, r’, t’). Two examples illustrate this. Firstly, the triplet (Nanjing, is the capital of, China) incorrectly associates Nanjing as the capital of China, whereas the correct head entity should be Beijing. Secondly, in the triplet (Jimmy Carter, is the son of, Chip Carter), the relationship between the two individuals, Jimmy Carter and Chip Carter, is inverted. In real-world knowledge graphs, there are no ground-truth datasets that include errors. To simulate noisy triplets, we introduce errors by randomly replacing either the head or tail of the original triplets. As a result, we obtain modified triplets with noise denoted as $S^{\prime }=\left(h^{\prime },r^{\prime },t^{\prime }\right)$. It is important to note that entities and relations outside of the knowledge graph are not considered in this process.*


**Definition** **2.**
***The triplet-level KG $\hat{G}$.** We consider a triplet (h,r,t) as a node, and we define a relationship between two triplets that share either the head entity or the tail entity, e.g., there is a link between $\left(e_{i},r_{ij},e_{j}\right)$ and $\left(e_{j},r_{jk},e_{k}\right)$. We denote $\hat{G}=\left(S,V,T\right)$, where S represents the set of triplets from the original knowledge graph G, V represents the edges connecting the triplets that share entities, and T represents the set of new triplets introduced in the process. As illustrated in Figure 1b, $X_{1}$ and $X_{3}$ share the entity Alzheimer’s disease, so they have a hidden connection.*


**Definition** **3.**
***The hyper-view KG $G^{\prime }$. ** For any given triplet (h,r,t) in G that shares the same (h,r) pair, we define $H_{t}=\left\{t_{i}|\left(h,r,t_{i}\right)\in S\right\}$ as the set of all tail entities associated with this pair. Similarly, we define $H_{h}=\left\{h_{i}|\left(h_{i},r,t\right)\in S\right\}$ as the set of all head entities that share the same relation r and tail t. As depicted in Figure 1a, the head entities at the top share the same couple (belong department, Neurology). Entities such as cerebral infarction, NeuroLyme disease, syringomyelia, and Alzheimer’s disease are likely to be labeled as diseases. Similarly, chestnut and ginkgo are likely to be labeled as food items.*


**Definition** **4.**
***Confidence score [19].** To classify a triplet, we introduce an estimated score ranging from 0 to 1. For a true triplet (h, r, t), the confidence score is expected to be close to 1, indicating a high degree of confidence in its accuracy.*


Unless otherwise specified, the boldface notation in the formulas represents vectors.

## 4. Methodology

Traditional methods for KG representation learning typically approach the problem by modeling the KG as a heterogeneous graph and learning embeddings for entities (nodes) and relations (edges) [14,26,27]. However, most algorithms struggle to handle the complex relations between triplets. To tackle this issue, several studies [32,33,34,35] propose methods that aggregate information from neighborhood nodes for latent messages. Furthermore, some studies [23,37] explore the integration of external information, such as entities’ labels, to enhance KG embedding. However, obtaining label information for entities in MKGs is not always a straightforward process. Thus, this paper proposes a novel framework named Enhancing error detection on Medical knowledge graphs via intrinsic labEL (EMKGEL). Inspired by recent work [22,23], we construct a triplet-level KG in Definition 2 to capture neighborhood information. In contrast to CAGED, which employs a hyper-parameter for error filtering, our approach involves extracting potential label information at a hyper-view level. We then integrate this information into the attention mechanisms within our framework. Our intuition is that a triplet can acquire analogous label feature information from the set associated with the same hyperedge. For example, the subclass words under the same parent-class words often share similar semantic contexts.

As illustrated in Figure 2, our proposed EMKGEL consists of a multi-view KG, a hyper-view GAT, and a joint confidence score. In our proposed approach, we first generate a triplet-level KG and a hyper-view KG, as described in Definitions 2 and 3. We then employ an attention mechanism to capture the intrinsic label information of entities in the hyper-view KG. This information is subsequently incorporated into the triplets using the hyper-view GAT to enhance the nodes. To train the model, we utilize a combination of KG embedding loss and global triplet embedding loss. This joint training enables the model to learn meaningful representations that capture both the structural relationships within the KG and the semantic contexts of the triplets. Finally, we estimate triplets using a joint confidence score based on the learned representations.

### 4.1. Hyper-View GAT for Representation Learning

Noisy triplets in KGs can have a detrimental effect on representation learning, thereby jeopardizing downstream tasks. It is crucial to ensure the reliability of the encoder to mitigate the impact of these errors. To address this challenge, CAGED introduces an error-aware GNN that filters out errors. However, determining an optimal hyper-parameter for error filtering is a complex task, making it challenging to apply CAGED to different KGs effectively. In this paper, we present a novel method to extract intrinsic label information from the hyper-view KG. We leverage this information to enhance the representation of triplets by incorporating it into a hyper-view GAT. By combining local structural information and neighborhood triplet messages, the hyper-view GAT effectively integrates multiple sources of information. Following the integration process, we estimate the reliability of each triplet by assigning a confidence score. This score serves as a measure of the triplet’s quality, aiding in the interpretation and utilization of the KG data.

**Figure 2 bioengineering-11-00225-f002:**
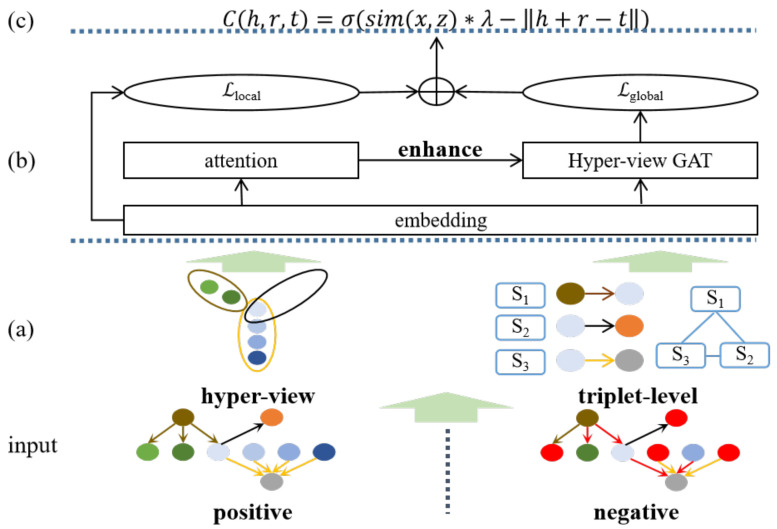
For training purposes, all triplets are generated with pairwise negative examples, denoted by red. This involves randomly replacing the head entity, the relation, and the tail entity in each triplet. (**a**) We construct a triplet-level KG and a hyper-view KG. (**b**) Enhanced by intrinsic label information, triplet-level nodes learn the embedding from neighborhood messages. (**c**) After training on a joint loss, we estimate the confidence of the combined local and global trustworthiness of triplets.

#### 4.1.1. Local Structural Information Modeling

Applying a GAT [42] for learning emphasizes the capture of latent information from neighborhood nodes. However, this approach can potentially weaken the inherent information contained within the triplets, denoted as h⟶r⟶t. Taking inspiration from CAGED, we utilize a bidirectional LSTM to acquire local representations that preserve the specific structural information of the triplets within global information learning.
(1)$$x=\left[x_{h};x_{r};x_{t}\right]=bi-lstm\left(h,r,t\right).$$

As shown in Equation (Equation 1), we initialize a triplet (h,r,t) and pass the resulting vector (h,r,t) through the bi-LSTM layer. The output x represents the local triplet embedding, which captures the structural information of the triplet. Subsequently, we utilize this local embedding x as the input for the global modeling layer.

#### 4.1.2. Intrinsic Label Information in Hyper-View KG

In the MKG, there is a wealth of untapped and exploitable unknown entity label information available. To uncover the latent label information of entities without external information, we construct a hyper-view KG that enriches each entity with diverse attributes within the medical domain. For an embedding triplet x, the head entity *h* and the tail entity *t* have corresponding sets, as defined in Definition 3. Taking the shared pair (h,r) as an example, we can obtain a set $H_{t}$ for the tail entity. Based on our assumption, the entities in $H_{t}$ likely share similar characteristics or properties, allowing us to infer potential labels or attributes for the tail entity based on this set.

To capture the messages associated with potential labels, we employ an attention mechanism,
(2)$$q_{t,i}=f_{att(t,t_{i})},$$$q_{t,i}$ indicates the coefficients across the original *t* and the i-th tail entity in $H_{t}$. $f_{att}\left(\cdot \right)$ denotes a single layer feed-forward neural network.

To reduce bias and ensure a fair comparison among the coefficients, we normalize them using a Softmax function. By incorporating information from the entire set $H_{t}$, we can capture the potential labels or attributes associated with $H_{t}$,
(3)$$\alpha_{t,i}=\frac{exp(q_{t,i})}{\sum_{j=1}^{H_{t}}exp\left(q_{t,j}\right)},$$
(4)$$t^{*}=\sum_{i=1}^{|H_{t}|}\alpha_{t,i}\cdot t_{i}.$$

Similarly, we obtain the $h^{*}$ from the all head entities of $H_{h}$,
(5)$$h^{*}=\sum_{i=1}^{|H_{h}|}\alpha_{h,i}\cdot h_{i}.$$

To estimate the importance of the $h_{i}^{*}$ and $t_{i}^{*}$, we employ the similarity calculation. The hyper-view score $hyper_{x}$ is utilized to gauge the significance of neighboring nodes,
(6)$$hyper_{x}=sim\left(\left(h,t\right),\left(h^{*},t^{*}\right)\right),$$By considering the hyper-view score, we can determine the level of contribution that the neighboring nodes have towards the representation and understanding of the target triplet.

#### 4.1.3. Neighborhood Information Modeling

Relying solely on KG embeddings to estimate the confidence of a triplet is insufficient. It is crucial to incorporate contextual information from neighboring triplets to enhance the confidence estimation process. While previous methods such as R-GCN [34] and KGAT [35] have shown effectiveness in leveraging neighborhood information, they may encounter a decline in performance when neighboring triplets contain noisy or erroneous messages, as highlighted in the work of CAGED [22]. To address this issue, CAGED introduces a hyper-parameter that helps mitigate the impact of errors. Given the abundant availability of entity label information in MKGs, we utilize hyper-view scores to enhance the neighborhood nodes.

Specifically, for a given anchor triplet x with m neighboring triplets $x_{1},x_{j},\cdots ,x_{m}$, we aggregate the information from these neighbors to update the representation of the anchor triplet. The weights of anchor triplet x and neighboring triplets $x_{j}$ are calculated as follows: (7)$$e_{j}=a\left(W_{c}x,W_{c}x_{j}\right),$$
where $e_{j}$ indicates the weight of $x_{j}$ to x. *a* indicates the attention function $a:R_{d}^{\prime }\times R_{d}^{\prime }⟶R$, $W_{c}\in R_{d}^{\prime }\times R_{d}^{\prime }⟶R$ is a trainable parameter matrix that projects triplet x into the same vector space.

Then, we incorporate the hyper-view score to enhance the label information, as depicted in Equations (Equation 2)–(Equation 6). In detail, we make a dot-product between $e_{j}$ and $hyper_{x_{j}}$ in normalization,
(8)$$\alpha_{j}=\frac{exp(hyper_{x_{j}}\cdot e_{j})}{\sum_{k=1}^{m}exp\left(hyper_{x_{k}}\cdot e_{k}\right)},$$
where $\alpha_{j}$ indicates the normalized coefficient weight of the j-th triplet $x_{j}$ to anchor triplet x.

Finally, we obtain the reconstructed vector z for the original anchor triplet x. And the σ(·) denotes the Sigmoid function,
(9)$$z=\sigma (\sum_{j=1}^{m}\alpha_{j}W_{c}x_{j}).$$

### 4.2. Joint Training Strategy

To capture the semantic and latent information, we introduce a training strategy to integrate KG embedding loss and global embedding loss.

Based on the translation assumption, we utilize the TransE score function $d_{local}$ to fit the local structural information,
(10)$$d_{local}=∥h+r-t∥.$$

For neighborhood information embedding, we use the $d_{global}$ to estimate the distance between the anchor triplet x and reconstruct triplet z as follows: (11)$$d_{global}=∥x-z∥.$$

To integrate the KG and global triplet embedding, we introduce the trade-off hyper-parameter λ. In Section 5, we specifically investigate the effect of different values of λ. The calculation is as follows: (12)$$d_{joint}=d_{local}+\lambda \cdot d_{global}.$$

Subsequently, we leverage a margin-based ranking loss function for negative sampling during the training process following the previous work [14],
(13)$$L=\sum_{(h,r,t)\in S}\sum_{\left(h^{\prime },r,t^{\prime }\right)\in S^{\prime }}{\left[\gamma +d_{joint}\left(h,r,t\right)-d_{joint}\left(h^{\prime },r,t^{\prime }\right)\right]}_{+},$$
where ${\left[\right]}_{+}$ equals max(0,x), λ is a hyper-parameter. *S* indicates the set of the original triplets, and $S^{\prime }$ indicates the set of the negative triplets that randomly replace the head and tail entities. It is crucial to ensure that corrupted triplets are not in *S* and are non-repetitive.

### 4.3. Confidence Score

After training, we obtain the confidence score, as depicted in Equation (Equation 14): (14)$$C\left(h,r,t\right)=\sigma (\lambda \cdot sim\left(x,z\right)-{∥h+r-t∥}_{2}^{2}),$$The σ(·) denotes the Sigmoid function. The function sim(·) denotes the similarity of the original triplet x and the reconstructed triplet z. The confidence score ranges from 0 to 1, with higher values indicating a stronger positive correlation for the triplet.

The learning process of our method is summarized in Algorithm 1.
**Algorithm 1 **Error detection on medical knowledge graphs via intrinsic label information**Input:** Knowledge graph G with noise**Output:** KG embeddings and confidence score
  1:Initialize network parameters,  2:Construct a triplet-level and a hyper-view KG as per Definitions 2 and 3, respectively.  3:**while** not converged **do**  4:   **for** each (h,r,t)∈ S **do**  5:   Modeling the local structural information of triplets as defined in Equation (Equation 1),  6:   Extract the intrinsic label information in hyper-view and then compute the importance score of triplets as defined in Equation (Equation 6),  7:   Acquire the representation in hyper-view GAT in Equation (Equation 9),  8:   Compute the KG embedding distance in Equation (Equation 10) and global triplet embedding distance in Equation (Equation 11). Combined with a trade-off parameter λ and obtain the joint loss in Equation (Equation 13).  9:**end while**10:Compute the confidence score as defined in Equation (Equation 14).


## 5. Experiments and Discussion

In this section, we will provide detailed experimental settings. Through the parameter analysis, ablation study, and case study, we validate the effectiveness of the proposed method, EMKGEL.

### 5.1. Experimental Settings

In this section, we provide a detailed overview of the experimental settings, including datasets, baseline methods, and evaluation metrics.

#### 5.1.1. Benchmark Datasets

Similar to prior studies, such as [19,22,23], we adopt the approach of randomly replacing head and tail entities to generate noisy triplets. As depicted in Definition 1, we introduce 5% noisy triplets into three medical real-world KGs and 5%, 10%, and 15% noisy triplets into one general KG to explore the robustness of our method.

PharmKG-8k [11] is a multi-relational attribute biomedical knowledge graph composed of more than 500,000 individual interconnections between genes, drugs, and diseases, with 29 relation types over a vocabulary of 8000 disambiguated entities.

DiaKG is derived from 41 publicly published diabetes guidelines and consensus documents, covering the most extensive range of research topics and hot areas in recent years.

DiseaseKG is a knowledge graph built upon common disease information utilizing the cnSchema framework.

Detailed information on the datasets is summarized in Table 2.

#### 5.1.2. Baseline Methods

In our experiments, we introduce KG embedding baseline methods and error detection baseline methods.

KG embedding: We compare them with the traditional representation learning methods, including TransE [14], DistMult [15], and RotatE [16]. We leverage the function score as the confidence score after training. In TransE, we employ the Euclidean distance ${-|h+r-t|}_{2}$ as the confidence score.

Error detection: For KG error detection methods, we evaluate the proposed method against state-of-the-art approaches, including CKRL [19], KGTtm [21], and CAGED [22].

#### 5.1.3. Evaluation Metrics

Consistent with previous studies [22,23], we adopt the practice of ranking all triplets based on their confidence score. Triplets with lower scores are considered potential candidates for being noisy triplets. Precision@K and Recall@K metrics are utilized to estimate the effectiveness. In detail, Precision@K denotes the TopK lowest confidence score triplets among the TopK triplets. Recall@K denotes the TopK lowest confidence score triplets among all triplets.
(15)$$Precision@K=\frac{\left|\mathrm{Errors}\;\mathrm{Discovered}\;\mathrm{in}\;\mathrm{TopK}\;\mathrm{Ranking}\;\mathrm{list}\right|}{K}$$
(16)$$Recall@K=\frac{\left|\mathrm{Errors}\;\mathrm{Discovered}\;\mathrm{in}\;\mathrm{TopK}\;\mathrm{Ranking}\;\mathrm{list}\right|}{\left|\mathrm{Total}\;\mathrm{Number}\;\mathrm{of}\;\mathrm{Errors}\;\mathrm{in}\;\mathrm{KG}\right|}$$

#### 5.1.4. Implementation Details

We conduct experiments on GPU NVIDIA GeForce RTX 3090; the Python version is 3.8 and the Pytorch version is 1.11.0. Based on the average in-degree of the datasets, the number of neighbors for each triplet in the four datasets is set to 59/2/7/2. The embedding hidden size is set to 100, the same as the bi-LSTM hidden size. Default Xavier initialization and an initial learning rate of 0.003 were used.

In our experiments, we explore different hyper-parameters to assess their impact on the results. The trade-off parameter λ is set from 0.001 to 1000, while the margin parameter γ is adjusted within the range of 0 to 1. To mitigate the impact of randomness introduced by erroneous triplets, we average the experimental data across 10 random seeds ranging from 0 to 9.

### 5.2. Results and Analysis

In this section, we conduct a comprehensive evaluation to assess the effectiveness of our method across four datasets. Through thorough observation and analysis, we demonstrate that our proposed method performs effectively in all four datasets. For clarity, we highlight the optimal results in black and underline the second-best results. Additionally, we indicate that Precision@K is equal to Recall@K when K equals ratios by using an asterisk (*).

#### 5.2.1. Main Results

As depicted in Table 3, the results demonstrate that (1) error detection methods outperform traditional embedding methods on the three medical KGs, and (2) notably, our proposed method, EMKGEL, outperforms all existing methods, delivering the best performance.

Specifically, traditional embedding methods such as TransE and RotatE solely focus on local structural information while neglecting global triplet embedding. Consequently, this limited perspective can result in losing important messages from neighborhood triplets. To tackle this issue, CAGED addresses the limitation by incorporating global triplet embedding through contrastive learning. This integration enables CAGED to capture and leverage the global context of triplets, detecting a higher number of errors in the KG. Different from CAGED’s utilization of the uncertainty parameter μ for error filtering, our approach takes a novel approach. Initially, we leveraged a hyper-view KG to extract potential label information for entities. Subsequently, we estimated the importance of nodes by assigning hyper-view scores, thereby enhancing their impact on neighboring nodes. In our experiments, we observe that our method demonstrates improvements of 0.7%, 6.1%, and 3.6% on PharmKG-8k, DiseaseKG, and DiaKG, respectively.

Additionally, we introduce different ratios of 5%, 10%, and 15% on Nell-995 to observe their effectiveness. As depicted in Table 4, our method consistently achieves the best results across different cases, highlighting the robustness of EMKGEL.

#### 5.2.2. Ablation Study

To validate the individual components of our proposed method, we conduct comprehensive experiments. Firstly, we replace the bi-LSTM by simply concatenating the triplet embedding to assess the impact of local structural information. As shown in Table 5, we can observe that the variant employing only concatenation exhibits inferior performance due to the absence of structural information. Secondly, we replace the hyper-view GAT with a simple GAT. As indicated in Table 5, our hyper-view GAT outperforms the GAT, thereby demonstrating the effectiveness of label information in enhancing the embeddings. Thirdly, we proceed to eliminate the KG and triplet embedding losses individually. Upon analyzing the results presented in Table 5, it becomes apparent that the model’s performance suffers when either the KG or triplet embedding loss is removed. This observation suggests a strong interdependence between KG embedding and triplet embedding, indicating that these two components work collaboratively to enhance the model’s performance. Lastly, we introduce a replacement for the TransE score function, as shown in Equation (Equation 10), by adopting the RotatE score function. This modification aims to explore the flexibility of different score functions and their impact on the model’s performance.

#### 5.2.3. Parameter Analysis

The λ is the trade-off parameter that balances the KG embedding (e.g., ∥h+r−t∥) and global triplet embedding (e.g., ∥x−z∥). To investigate the impact of λ, we set it from 0.001 to 1000. We conducted experiments on all four datasets and the evaluation on Recall@K. Based on the findings depicted in Figure 3a, we observe that (1) PharmKG-8k, DiaKG, and Nell-995 demonstrate their best performance when the value of λ is set to 10; (2) DiseaseKG achieves its optimal result when λ is set to 0.1; (3) at the outset, as λ increases, the model’s performance shows improvement. However, once it reaches the optimal value, a decline in performance is observed. This suggests that while enhancing the impact of global embedding initially boosts performance, there is a point of diminishing returns. Pushing the value of λ beyond this point does not yield the best performance for the model.

The γ is the margin parameter. As shown in Figure 3b, the optimal result is γ=0.5, and the trends of the four datasets are essentially identical.

### 5.3. Case Study

To investigate how the hyper-score enhances error detection, we conducted a case study on the anchor triplet (Diabetes ketoacidosis, Symptom, Polydipsia) of DiseaseKG. As shown in Figure 4, we present the hyper-scores of neighbors. In our assumption, the hyper-view GAT will enhance the triplets with high hyper-scores and reduce the impact of triplets with low hyper-scores. Finally, we assume that the confidence score will significantly differentiate between true triples and noisy triples.

To confirm the results, we set the number of share(h, r)/share(r, t) of $X_{1}$ to $X_{5}$, and the confidence score of each triplet, respectively.

As shown in Table 6, we observe that our proposed method outperforms CAGED. In detail, CAGED cannot lower the confidence score on three noisy triplets, while ours does so and to a better degree. Furthermore, $X_{2}$, $X_{5}$, and $X_{3}$ possess more intrinsic label information, as indicated by their hyper-scores of {0.8993, 0.9103, 0.5371}, due to a high number of share(h, r) and share(r, t). Hyper-view GAT enhances the representation of $X_{2}$, $X_{5}$, and $X_{3}$. Conversely, $X_{1}$ and $X_{4}$ contain less intrinsic information as they have a low number of share(h, r) and share(r, t). Consequently, hyper-view GAT diminishes the influence of $X_{1}$ and $X_{4}$. However, the confidence score of noisy triplet $X_{3}$ (Diabetes ketoacidosis, Medication, Shagliptin tablets) in both methods is ambiguous. In real-world scenarios, it is not enough to prescribe medication based solely on simple triplet information; it is also necessary to consider the actual situation. Therefore, this is a common issue with existing methods based on knowledge graphs.

To show a straightforward validation of our method, we present a visualization in Figure 5. The y-axis represents the confidence scores assigned to the triplets. True triplets are denoted by green nodes, while false triplets are represented by red nodes. In comparison to CAGED in Figure 5b, our model in Figure 5a assigns lower confidence scores to noisy triplets, approaching zero. This visualization serves as evidence of the effectiveness of our model in real-world scenarios.

## 6. Conclusions

In this paper, we propose a novel framework named Enhancing error detection on Medical knowledge graphs via intrinsic labEL (EMKGEL). Firstly, we construct a hyper-view KG and a triplet-level KG. The former aims to capture intrinsic label information, and the latter focuses on neighboring information. Secondly, we introduce the hyper-view GAT to incorporate the entity label information into the triplet. Then, we integrate KG embedding and global triplet embedding in the training stage. In the end, we estimate each triplet by their confidence score. The evaluation on three medical KGs and one general KG demonstrates the effectiveness of EMKGEL. We believe that our method can be an effective tool for error detection during KG construction.

However, there remain some challenges that future work needs to address:Existing error detection methods [19,21,22,23] only take into account the entities and relations already present in the knowledge graph, while error triplets could originate from outside the dataset [43]. To address the limitation, combining textual information from large language models with graph structure information is a promising direction [44,45].The currently used max dataset size is approximately 500,000, and in the future, we will consider conducting more experiments on large-scale graphs while maintaining the effectiveness and reducing the training time.We will explore more meaningful downstream tasks, such as knowledge-based question answering in the medical field.

## Figures and Tables

**Figure 1 bioengineering-11-00225-f001:**
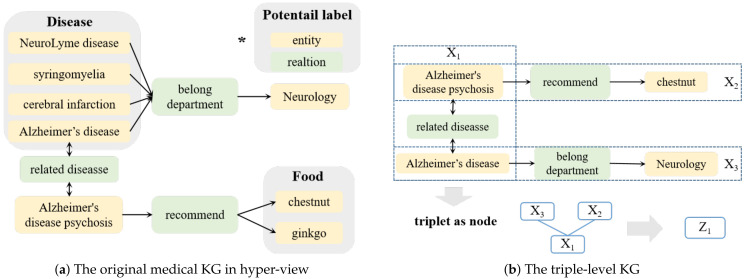
The medical KG in hyper-view and triplet-level. (**a**) The hyper-view KG. * represents the example template: entities are represented by yellow, while relations are denoted by green. Within the hyper-view, potential label information is highlighted in black. Specifically, tail entities sharing the same couple (Alzheimer’s disease psychosis, recommend) have a hidden potential label as food. Head entities sharing the same couple (belong department, neurology) have a potential label as disease. (**b**) The triplet-level KG. We use $X_{1}$, $X_{2}$, $X_{3}$ to represent triplets. Triplets’ shared entities have hidden connections. Aggregating the neighborhood triplets $X_{2}$, $X_{3}$ of the triplet $X_{1}$, we reconstruct it as $Z_{1}$.

**Figure 3 bioengineering-11-00225-f003:**
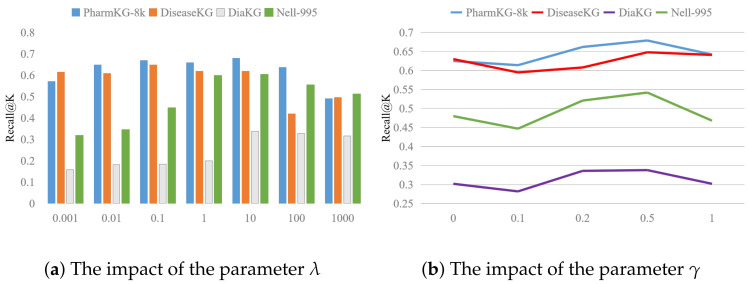
Impact of hyper-parameters on the four datasets.

**Figure 4 bioengineering-11-00225-f004:**
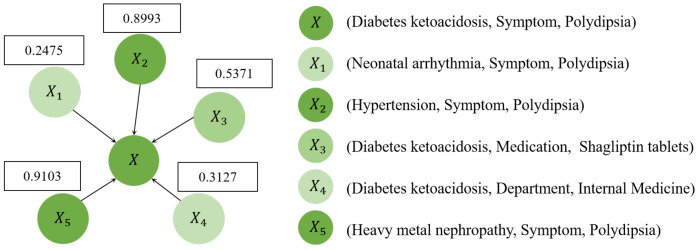
Case study on DiseaseKG. The center triplet of subgraph is (Diabetes ketoacidosis, Symptom, Polydipsia). The other triplets are neighbors of the anchor triplets, which share the same head entity or tail entity, as defined in Definition 2. Detailed triplets are provided on the right side.

**Figure 5 bioengineering-11-00225-f005:**
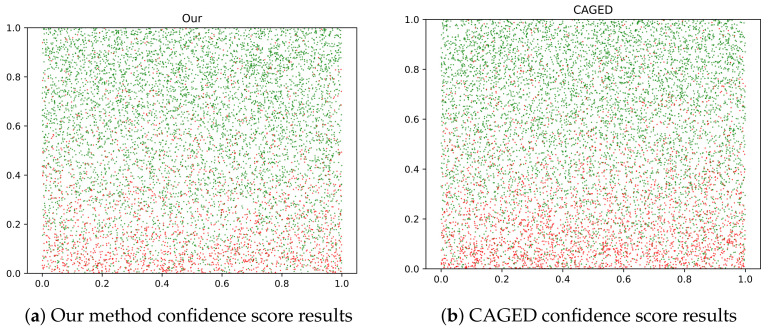
Scatter of confidence score on DiseaseKG.The red nodes denote noisy triplets and green nodes denote true triplets.

**Table 1 bioengineering-11-00225-t001:** Major notations.

Notation	Description
G	the original knowledge graph
(h, r, t)	a triplet with the head, relation, and tail
$\hat{G}$	the triplet-level KG constructed from G
$G^{\prime }$	the hyper-view KG constructed from G
$H_{t}$	the set of the tail entities share the same (h, r)
**x**	representation of the triplet
**z**	representation learned from x
C(h,r,t)	confidence score of the triplet (h, r, t)

**Table 2 bioengineering-11-00225-t002:** The statistical information of datasets.

Datasets	Entities	Relations	Triplets	In-Degree
PharmKG-8k	7262	28	500,958	58.86
DiaKG	2658	15	4099	1.55
DiseaseKG	43,972	12	312,159	6.74
Nell-995	75,492	200	154,213	1.98

**Table 3 bioengineering-11-00225-t003:** Error detection results on three datasets with noisy ratio = 5%. Optimal results are highlighted in black, while the second-best results are underlined. † denotes the result is reproduced by ourselves. * denotes the Precision@K is equal to Recall@K.

	Datasets	PharmKG-8k					DiseaseKG					DiaKG				
	**K**	**K = 1%**	**K = 2%**	**K = 3%**	**K = 4%**	**K = 5%** *****	**K = 1%**	**K = 2%**	**K = 3%**	**K = 4%**	**K = 5% ***	**K = 1%**	**K = 2%**	**K = 3%**	**K = 4%**	**K = 5% ***
Precision@K	TransE [14] †	0.812	0.681	0.584	0.515	0.461	0.623	0.498	0.425	0.370	0.331	0.341	0.280	0.227	0.207	0.175
	DistMult [15] †	0.802	0.722	0.664	0.607	0.547	0.754	0.636	0.533	0.458	0.402	0.439	0.304	0.252	0.219	0.200
	RotatE [16] †	0.871	0.762	0.681	0.609	0.563	0.727	0.601	0.501	0.427	0.374	0.487	0.317	0.260	0.225	0.195
	CAGED [22] †	0.958	0.917	0.845	0.761	0.674	0.936	0.870	0.776	0.687	0.611	0.562	0.459	0.400	0.371	0.326
	EMKGEL(our)	**0.963**	**0.923**	**0.852**	**0.763**	**0.679**	**0.962**	**0.899**	**0.813**	**0.727**	**0.648**	**0.596**	**0.487**	**0.418**	**0.378**	**0.338**
Recall@K	TransE [14] †	0.162	0.272	0.350	0.411	0.461	0.124	0.199	0.255	0.296	0.331	0.068	0.112	0.136	0.165	0.175
	DistMult [15] †	0.160	0.288	0.398	0.486	0.547	0.150	0.254	0.319	0.367	0.402	0.087	0.122	0.151	0.175	0.200
	RotatE [16] †	0.174	0.304	0.408	0.487	0.563	0.142	0.241	0.301	0.342	0.374	0.097	0.126	0.156	0.180	0.195
	CAGED [22] †	0.191	0.366	0.507	0.608	0.674	0.187	0.347	0.465	0.550	0.611	0.112	0.183	0.240	0.297	0.326
	EMKGEL(our)	**0.192**	**0.369**	**0.511**	**0.611**	**0.679**	**0.192**	**0.359**	**0.487**	**0.581**	**0.648**	**0.119**	**0.194**	**0.251**	**0.302**	**0.338**

**Table 4 bioengineering-11-00225-t004:** Error detection results on Nell-995 with different ratios. Optimal results are highlighted in black, while the second-best results are underlined. † denotes the result is reproduced by ourselves.

Ratio	5%			10%			15%		
**K**	**K = 5%**	**K = 10%**	**K = 15%**	**K = 5%**	**K = 10%**	**K = 15%**	**K = 5%**	**K = 10%**	**K = 15%**
	Precision@K								
TransE [14] †	0.358	0.255	0.196	0.546	0.432	0.356	0.666	0.569	0.490
DistMult [15] †	0.286	0.235	0.205	0.489	0.421	0.375	0.605	0.540	0.494
RotatE [16] †	0.352	0.237	0.193	0.454	0.378	0.324	0.626	0.544	0.474
CKRL [22]	0.450	0.306	0.236	0.679	0.524	0.421	0.745	0.646	0.560
KGTtm [22]	0.481	0.320	0.242	0.713	0.527	0.437	0.788	0.673	0.576
CAGED [22]	0.516	0.325	0.251	0.799	0.585	0.458	0.823	0.729	0.599
EMKGEL(our)	**0.542**	**0.344**	**0.264**	**0.807**	**0.606**	**0.488**	**0.842**	**0.732**	**0.626**
	Recall@K								
TransE [14] †	0.358	0.511	0.589	0.273	0.432	0.534	0.222	0.379	0.490
DistMult [15] †	0.286	0.471	0.616	0.244	0.421	0.562	0.201	0.360	0.494
RotatE [16] †	0.352	0.474	0.579	0.227	0.378	0.487	0.208	0.363	0.474
CKRL [22]	0.450	0.612	0.708	0.340	0.524	0.632	0.248	0.431	0.560
KGTtm [22]	0.481	0.640	0.726	0.357	0.527	0.656	0.263	0.449	0.576
CAGED [22]	0.516	0.650	0.753	0.400	0.585	0.687	0.274	0.486	0.599
EMKGEL(our)	**0.542**	**0.689**	**0.792**	**0.403**	**0.606**	**0.732**	**0.281**	**0.488**	**0.732**

**Table 5 bioengineering-11-00225-t005:** Ablation study on Nell-995 with noisy Ratio = 5%. Optimal results are highlighted in black.

	Precision@K					Recall@K				
**K**	**K = 1%**	**K = 2%**	**K = 3%**	**K = 4%**	**K = 5%**	**K = 1%**	**K = 2%**	**K = 3%**	**K = 4%**	**K = 5%**
EMKGEL	**0.887**	**0.792**	**0.692**	**0.613**	**0.542**	**0.177**	**0.317**	**0.415**	**0.490**	**0.542**
EMKGEL w/o bi-LSTM	0.793	0.711	0.621	0.549	0.492	0.158	0.284	0.373	0.439	0.492
EMKGEL w/o HyGAT	0.798	0.721	0.630	0.561	0.507	0.159	0.288	0.378	0.448	0.507
EMKGEL w/o local	0.866	0.759	0.658	0.574	0.507	0.173	0.303	0.394	0.459	0.507
EMKGEL w/o global	0.828	0.754	0.677	0.594	0.518	0.165	0.301	0.406	0.475	0.518
EMKGEL w RotatE	0.885	0.768	0.661	0.583	0.523	0.177	0.307	0.397	0.467	0.523

**Table 6 bioengineering-11-00225-t006:** The information of neighbors of the anchor triplet. The red color denotes bad case.

Triplet	Number of Share(h, r)	Number of Share(r, t)	Confidence Score		
			**Grounded**	**Our**	**CAGED**
*X*	66	9	True	0.8703	0.8549
$X_{2}$	66	9	True	0.8301	0.7924
$X_{5}$	66	11	True	0.9201	0.8773
$X_{3}$	66	10	Noisy	0.6521	0.4891
$X_{1}$	5	2	Noisy	0.2431	0.6312
$X_{4}$	4	1	Noisy	0.1003	0.5455

## Data Availability

The datasets we used in experiments are available. PharmKG-8K (https://github.com/biomed-AI/PharmKG, accessed on 10 December 2023), DiseaseKG (http://openkg.cn/dataset/disease-information, accessed on 10 December 2023), DiaKG (http://openkg.cn/dataset/diakg, accessed on 10 December 2023), Nell-995 (https://paperswithcode.com/dataset/nell-995, accessed on 10 December 2023).

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
