# Peer review of "Enhancing Error Detection on Medical Knowledge Graphs via Intrinsic Label"

_bioengineering, 2024, doi:10.3390/bioengineering11030225_

Round 1
Reviewer 1 Report
Comments and Suggestions for Authors
Comment
The paper presents a method for improving error detection on medical knowledge graphs via intrinsic labels. The authors improve the confidence score by combining both local and global trustworthiness to calculate the triplets. The results from the evaluations on three data sets show a marginal improvement in comparison with existing methods, such CAGED, RotaE, etc.
Overall, I believe that the technical contribution of the paper is solid, and the work is an incremental contribution to the field. The paper’s writing and presentation are adequate and easy to follow. The background, related work and methodology are adequate. In my opinion, the paper can be accepted after the improvement of the minor points:
Section 5.3 Case study: this section is too brief, and it needs to be improved and provide further explanation and description. It is almost impossible to compare the advantage(s) of each method in the two scatterplots in Figure 4. To support the claim, additional analytical methods and/or enhancement in the visualisation are required. Also, as this is part of a comparison, you need to indicate the background of the datasets in this visualisation. The section heading Cast study could also mislead if this is a new case study, or this is just a visualisation of existing “case studies” of the three datasets? Etc. Finally, it would be more convincing if the authors could provide additional case studies to illustrate the effectiveness of the proposed method on different samples.
Section 6 Discussion may also need to be improved. If it is a discussion, it should discuss the finding by providing more arguments and possible explanation(s) of the outcome. In the current form, the section is more about a limitation and future work. If this is the case, the authors can consider merging this section into the Conclusion section.
Comments on the Quality of English Language
The english language is generally good. It could be more beneficial if the authors can do another improvement in the proof reading, especially in the expression.
Author Response
Dear Reviewer,
Thank you very much for taking the time to review this manuscript. Please find the detailed responses below and the corresponding revisions/corrections highlighted/in track changes in the re-submitted files.
Comments 1: Section 5.3 Case study: this section is too brief, and it needs to be improved and provide further explanation and description.
Response 1: Agree. We supplemented this section by taking the triplets (Diabetes ketoacidosis, Symptom, Polydipsia) from DiseaseKG as an example, investigating the hyper scores of their neighboring triplets, and comparing the confidence score with the CAGED method. We observe that triplet with a high number of share(h, r)/share(r, t) may get a higher hyper score. Finally, due to the hyper score, Hyper-view GAT enhances the true triplets and reduces the impact of noisy triplets, our method outperforms CAGED. Please refer to Table 6 and Figure 4 in Section 5.3 Case study for specific details.
Comments 2: Discussion may also need to be improved.
Response 2: Agree. In fact, in section 5 Experiments and Discussion, we have already analyzed and discussed the results as well as the proposed method. Therefore, we remove the discussion section and merge the existing content into section 6 Conclusion.
Response 3: Additional clarifications
We added Table 1 Major notations in Section 3.
We replaced some formulas and symbols with highlighted.
We cited each figure, table, and equation.
We checked the language.
I hope my response will be helpful to you!
Reviewer 2 Report
Comments and Suggestions for Authors
The article entitled as "Enhancing Error Detection on Medical Knowledge Graphs via Intrinsic Label" is well presented and contains significant contributions for the journal. This work will be further imrpoved after the inclusion of following comments:
1. Author has to define each and every acronyms and abbreviations before use it.
2. Only few equations are properly cited in text. Therefore, author has to cite each figure, table and equations at suitable place.
3. As author used only two parametrs for experiment performance evalution. Author has to also perform some statistical analysis also.
4. Author has to elaborate section 5.3 i.e., case study.
5. Author has to perform comparative analysis of performances between proposed model performances and previous work performance.
6. Minor language corrections required.
Comments on the Quality of English LanguageThere are few english mistakes. Author has to check it throughly and make corrections if required.
Author Response
Dear Reviewer,
Thank you very much for taking the time to review this manuscript. Please find the detailed responses below and the corresponding revisions/corrections highlighted/in track changes in the re-submitted files.
Comments 1: Author has to define each and every acronyms and abbreviations before use it.
Response 1: Agree. We corrected these issues and highlighted them. We also added Table 1 "Major notations" in Section 3.
Comments 2: Only few equations are properly cited in text. Therefore, author has to cite each figure, table and equations at suitable place.
Response 2: We cited the each figure, table and equations.
Comments 3: As author used only two parametrs for experiment performance evalution. Author has to also perform some statistical analysis also.
Response 3: In the existing methods CAGED and AKAE, only precision@K and recall@K metrics are used for evaluation on error detection. However, to investigate the performance of intrinsic label information, we statistic the number of share(h,r) and share(r,t) as shown in Table 6 in the 5.3 case study.
Comments 4: Author has to elaborate section 5.3 i.e., case study.
Response 4: Agree. We supplemented this section by taking the triplets (Diabetes ketoacidosis, Symptom, Polydipsia) from DiseaseKG as an example, investigating the hyper scores of their neighboring triplets, and comparing the confidence score with the CAGED method. We observe that triplet with a high number of share(h, r)/share(r, t) may get a higher hyper score. Finally, due to the hyper score, Hyper-view GAT enhances the true triplets and reduces the impact of noisy triplets, our method outperforms CAGED. Please refer to Table 6 and Figure 4 in Section 5.3 Case study for specific details.
Comments 5: Author has to perform comparative analysis of performances between the proposed model performances and previous work performance.
Response 5: In section 5.2.1 Main Results, we have analyzed the performance between our proposed method and previous work. In conclusion, (1)error detection methods outperform traditional embedding methods on the three medical KGs. (2)our proposed method outperforms all existing methods. Reasons for conclusion 1, embedding methods lose the neighborhood information. Reason for conclusion 2, compared to the baseline method CAGED, our method enhances the triplet by intrinsic information rather than filtering triplets by a hyperparameter. Furthermore, we did a case study to investigate the performance of our method.
Response 6: Additional clarifications
In fact, in section 5 Experiments and Discussion, we already analyzed and discussed the results as well as the proposed method. Therefore, we removed the discussion section and merged the existing content into section 6 Conclusion.
I hope my response will be helpful to you!